# Wind Turbine Blade Fault Detection Method Based on TROA-SVM

**DOI:** 10.3390/s25030720

**Published:** 2025-01-24

**Authors:** Zhuo Lei, Haijun Lin, Xudong Tang, Yong Xiong, He Wen

**Affiliations:** 1College of Engineering and Design, Hunan Normal University, Changsha 410081, China; leizhuo_0811@163.com (Z.L.);; 2College of Computer Science and Electronic Engineering, Hunan University Changsha 410082, China; 3Hunan Lianzhi Technology Co., Ltd., Changsha 410200, China; 4College of Electrical and Information Engineering, Hunan University Changsha 410082, China

**Keywords:** wind turbine blades, feature extraction, fault diagnosis, tyrannosaurus optimization algorithm, support vector machine

## Abstract

Wind turbines are predominantly situated in remote, high-altitude regions, where they face a myriad of harsh environmental conditions. Factors such as high humidity, strong gusts, lightning strikes, and heavy snowfall significantly increase the vulnerability of turbine blades to fatigue damage. This susceptibility poses serious risks to the normal operation and longevity of the turbines, necessitating effective monitoring and maintenance strategies. In response to these challenges, this paper proposes a novel fault detection method specifically designed for analyzing wind turbine blade noise signals. This method integrates the Tyrannosaurus Optimization Algorithm (TROA) with a support vector machine (SVM), aiming to enhance the accuracy and reliability of fault detection. The process begins with the careful preprocessing of raw noise signals collected from wind turbines during actual operational conditions. The method extracts vital features from three key perspectives: the time domain, frequency domain, and cepstral domain. By constructing a comprehensive feature matrix that encapsulates multi-dimensional characteristics, the approach ensures that all relevant information is captured. Rigorous analysis and feature selection are subsequently conducted to eliminate redundant data, thereby focusing on retaining the most significant features for classification. A TROA-SVM classification model is then developed to effectively identify the faults of the turbine blades. The performance of this method is validated through extensive experiments, which indicate that the recognition accuracy rate is 98.7%. This accuracy is higher than that of the traditional methods, such as SVM, K-Nearest Neighbors (KNN), and random forest, demonstrating the proposed method’s superiority and effectiveness.

## 1. Introduction

Wind energy [1,2], recognized as a sustainable and environmentally friendly energy source, has increasingly solidified its role within the global energy mix in recent years. The rapid expansion of the wind power industry has not only spurred advancements in renewable energy technologies [3], but has also significantly contributed to the evolution of wind power equipment. This includes notable innovations and the scaling up of wind turbine designs, which are crucial for enhancing energy generation capacity. However, as the number of wind power installations continues to rise and operational hours accumulate, the issue of wind turbine failures has become increasingly prominent. Frequent malfunctions can lead to safety incidents, posing risks to personnel and resulting in substantial economic losses [4]. These problems have been critical factors that have hindered the further advancement of the wind power sector. In particular, within complex natural environments, the health of wind turbine blades, which is one of the core components of wind power systems, directly influences the operational efficiency and safety of the entire wind farm. As such, accurately detecting the failure states of wind turbine blades is of paramount importance [5]. Effective fault detection [6] not only enables the timely identification and resolution of potential safety hazards, but also minimizes downtime and the economic repercussions associated with unexpected failures. Moreover, improving the reliability and operational efficiency of wind turbine equipment plays a crucial role in fostering the sustainable development of the wind power industry. By implementing advanced monitoring and maintenance strategies, stakeholders can ensure the longevity and performance of wind turbines, thereby maximizing energy production while safeguarding both human resources and financial investments. In conclusion, addressing the challenges of wind turbine reliability through innovative detection methods is essential for unlocking the full potential of wind energy and ensuring its viability as a key player in the global energy landscape.

In the field of wind turbine blade fault detection, current research focuses on the utilization of a variety of physical phenomena [7], such as blade-induced vibration characteristics [8], acoustic emissions [9], ultrasonic waves [10], and bearings [11,12], and thermal imaging [13] as sources of diagnostic information. It also looks at the output power difference, blade resonance frequency, and other external environment and internal operating conditions for diagnosis, and explores the use of advanced computer vision technology for image recognition [14,15] to reveal potential blade faults. However, each of these methods have their own problems. For example, monitoring of physical signals is often accompanied by the need for additional sensors, which not only adds cost but can also be tricky due to installation complexity and potential risks, and in-depth analysis of these signals requires a high degree of expertise, further complicating the diagnostic process. The image detection technology [16], although intuitive, is susceptible to external conditions, such as changing light and weather conditions, which can weaken the accuracy of the detection. At the same time, the high computational demands of image processing limit the feasibility of real-time analysis.

In view of these limitations, acoustic signals are easier to obtain, have the advantages of high efficiency, low cost, and being very easy to install, and are now widely used in the identification of mechanical faults. In [17], a sound recognition method based on Mel’s logarithmic spectrum and convolutional neural network is proposed to construct a fault classification model to successfully recognize transformer discharge and mechanical fault sound. In [18], an improved local mean decomposition method is proposed to realize composite fault diagnosis of gearboxes by separating the FM-AM components in the sound signals. In [19], a periodic audio cutting method based on clustering and median convergence is proposed to effectively cut and analyze the sound patterns, which can effectively identify the anomalies present in the blades.

These above methods have achieved good results in the field of audio signal pattern classification, which provides a reference for wind turbine blade fault detection. In this paper, a multi-scale fusion audio feature analysis [20] is added to enhance the wind turbine blade fault detection method. In this method, the features of audio signals are extracted in the time domain, frequency domain, and cepstrum domain [21], and then a multivariate feature matrix is constructed. Subsequently, the support vector machine (SVM) [22,23] algorithm is used for fault classification and recognition. At the same time, in order to optimize the SVM parameters, the Tyrannosaurus Rex Optimization Algorithm (TROA) [24] is applied, which simulates the search and attack strategies of Tyrannosaurus Rex during hunting, iteratively searches with the global optimal solution, and then obtains the optimal values of SVM model parameters to improve the efficiency and accuracy of wind turbine fault detection. The comparative analysis of experimental results confirms the accuracy and practicality of this proposed wind turbine blade fault detection method.

## 2. Method for Wind Turbine Blade Fault Detection Based on TROA-SVM

### 2.1. Fault Detection Framework for Wind Turbine Blades

Under the influence of external wind, wind turbines generate torque through their internal wind guiding mechanisms and blades, converting wind energy into mechanical energy. However, due to various influences, including wind conditions, mechanical factors, and operational environments, wind turbines are susceptible to failures during work. In this paper, we propose a novel approach utilizing acoustic signal acquisition equipment strategically positioned at various measurement points on the wind turbine to accurately and effectively collect sound signals from the blades. Following data collection, the fault features are extracted by using three analysis methods, i.e., the time domain analysis, time–frequency domain analysis, and cepstral domain analysis. These extracted features are then input into a machine learning model for assessing the working state of the blades and effectively detect any faults, as shown in Figure 1.

### 2.2. Multi-Scale Feature Extraction

The causes of faults in wind turbine blades are inherently complex and often challenging to identify. Feature extraction is a prerequisite for fault detection. The original noise signal features from the time domain, frequency domain, and cepstral domain are separately extracted, and then the faults are detected by the TROA-SVM model.

#### 2.2.1. Time Domain Characteristics

(1)Short-time energy

The short-time energy is a crucial parameter for assessing the strength of an audio signal, as it effectively reflects variations in signal amplitude. By capturing the fluctuations in energy magnitude within each time frame, short-time energy provides valuable insights into the dynamics of the signal. This parameter plays a significant role in various applications, enhancing our understanding of sound characteristics and improving the effectiveness of audio-related technologies. The short-time energy is defined by as follows:(1)$$E(i)=\overset{L-1}{\underset{n=0}{\sum }}y_{i}^{2}(n)$$
where E is the short-time energy, $y_{i}(n)$ is the i frame audio signal, and L is the frame length.

(2)Short-time average zero crossing rate

The short-time average zero crossing rate is a key characteristic in the analysis of time domain audio signals. Mathematically, it is defined as the number that the waveform crosses the zero point within each frame of the signal. Due to its ability to provide insight into signal clarity, the zero crossing rate has become an important feature in audio signal processing, enhancing the accuracy of sound classification and analysis. The short-time average zero crossing rate is defined by(2)$$Z(i)=\frac{1}{2}\overset{L-1}{\underset{n=0}{\sum }}\left|sgn\left[y_{i}(n)\right]-sgn\left[y_{i}(n-1)\right]\right|$$
where $y_{i}(n)$ is the signal amplitude at the *n* the point of the audio of the i frame, L is the frame length, and sgn[] is the symbol function.

#### 2.2.2. Frequency Domain Characteristics

The short-time power spectral density is a key frequency domain characteristic that reveals the relationship between a signal’s variance and its frequency by converting the time domain signal into the frequency domain. It is defined as the signal’s power distributed across various frequency bands. By analyzing short-time power spectral density, one can observe how the signal fluctuates within different frequency ranges and identify the bands where the most significant variations occur. This method enriches the analysis by providing more comprehensive information about the signal’s behavior in the frequency domain, which is an indispensable tool in audio and signal processing.

#### 2.2.3. Cepstral Domain Characteristics

The Mel Frequency Cepstral Coefficient (MFCC) [25] is a method based on the principles of the human auditory system, and it is widely used for analyzing audio signals. It simulates the critical bandwidth characteristics of the human ear by designing a set of triangular bandpass filter banks arranged from sparse to dense, covering the range from low frequency to high frequency. Usually, the number of filter banks is similar to the number of critical bands. In practice, the number of filter banks H is 22–26. In this paper, H = 26. The MFCC triangular filter is shown in Figure 2.

The process of MFCC feature extraction is as follows:(1)Preprocessing. The preprocessing step involves three processes, i.e., pre-emphasis, frame-splitting, and windowing. The pre-emphasis is to enhance high-frequency components of the signal. Next, the signal is divided into frames, i.e., the frame splitting, which is used to capture the short-term characteristics of the audio signal. Finally, windowing is applied to each frame to reduce signal discontinuities at the boundaries. By smoothing the edges of the frames, windowing minimizes abrupt transitions between frames, ensuring more accurate feature extraction.(2)Fourier transform: The time domain signal x(n) is converted to the frequency domain signal X(k) to obtain the linear power spectrum of the signal, i.e.,(3)$$X(k)=\overset{N-1}{\underset{n=0}{\sum }}x(n)\cdot e^{-2j\pi kn/N}$$(3)Calculating the logarithmic energy spectrum. The square of the frequency domain signal *X*(*k*) is calculated to obtain the energy spectrum, then multiple Mel filters are used for filtering and logarithmic operations are performed, i.e.,
(4)$$E(m)=\ln\left(\overset{N-1}{\underset{k=0}{\sum }}H_{m}(k)\cdot {\left|X(k)\right|}^{2}\right),\;0\leq m<M$$
where E(m) is the logarithmic energy calculated after the Mel filter, $H_{m}(k)$ is the Mel filter, and N is the number of Mel filters.
(4)Discrete cosine transform. The MFCC parameters are obtained by using the discrete cosine, i.e.,
(5)$$C(n)=\overset{M-1}{\underset{m=0}{\sum }}E(m)\cdot \cos\left(\frac{n\pi \left(m-\frac{1}{2}\right)}{M}\right),\;0\leq m<M$$
where C(n) is the MFCC parameters.



### 2.3. TROA-SVM Model Design

After preprocessing and feature extraction of sound data, a support vector machine (SVM) for fault classification is proposed to evaluate the operating status of wind turbine blades. SVM is particularly suitable for processing small-sample, nonlinear, and high-dimensional data problems, which can adapt well to wind turbine blade fault detection. However, SVMs are very sensitive to parameters, and in order to obtain the best model, their parameters need to be optimized, including the penalty coefficient, *C,* and the kernel function parameter gamma. In order to optimize *C* and gamma, the Tyrannosaurus Rex Optimization Algorithm (TROA) is employed to search and optimize SVM’s parameters, in order to find the optimal parameter combination and improve the performance of the SVM model.

TROA is a heuristic optimization algorithm based on the hunting behavior of Tyrannosaurus Rex. This algorithm simulates the search and attack strategies of Tyrannosaurus Rex during hunting, and iteratively searches for the global optimal solution. The algorithm first initializes a set of random solutions to simulate the search behavior of the Tyrannosaurus Rex at different locations. In each iteration, the position of the solution is adjusted based on the distance and direction between the current solution and the global optimal solution, simulating the approach and hunting behavior of the Tyrannosaurus Rex. This method effectively combines global search and local development, and can gradually approach the global optimum solutions, thereby achieving the optimal parameters of the SVM model. The TROA-SVM algorithm is described as follows:
(1)Initialize the positions of the Tyrannosaurus Rex population $X_{i}$, where each position represents a parameter vector
$\left(C_{i},{\mathrm{gamma}}_{i}\right)$, i.e.,
(6)$$X_{i}=(C_{i},{\mathrm{gamma}}_{i})$$
where *i* = 1, 2, …, *N*.(2)Update position and velocity of the Tyrannosaurus Rex. The position and velocity of each Tyrannosaurus Rex are updated based on the inertial weights w, the acceleration constants $c_{1}$ and $c_{2}$, and the individual best position $P_{best}$ and the global best position $G_{best}$, i.e.,
(7)$$V_{i}(t+1)=w\cdot V_{i}(t)+c_{1}\cdot rand\cdot (P_{best}-X_{i}(t))+c_{2}\cdot rand\cdot (G_{best}-X_{i}(t))$$
(8)$$X_{i}(t+1)=X_{i}(t)+V_{i}(t+1)$$

In order to avoid falling into a local optimum, a randomized perturbation is performed; the intensity of the perturbation is α and randn is a random number that follows a standard normal distribution, i.e.,(9)$$X_{i}(t+1)=X_{i}(t+1)+\alpha \cdot randn$$

These steps are repeated until the maximum number of iterations is achieved or a satisfactory classification accuracy is obtained. The finally obtained global optimal position $G_{best}$ is the optimized SVM parameter (C,gamma). In addition, the Tyrannosaurus Rex optimization algorithm also introduces an adaptive mechanism to dynamically adjust the search strategy based on the distribution of solutions and search progress during the iteration process. For example, when the iteration speed is slow during solving, the algorithm will increase the breadth of the search to avoid getting stuck in local optima. On the contrary, when the solution approaches the global optimal solution, the algorithm will increase the refinement of local iterations to improve search accuracy. This adaptive mechanism further enhances the robustness and search efficiency of the algorithm. By using these strategies, the Tyrannosaurus optimization algorithm can effectively optimize the parameters of the SVM, and improve the performance of the classification model. In wind turbine fault detection applications, this optimization process ensures that the model is able to accurately capture fault features and provide reliable health state assessment results. The fault detection algorithm based on TROA-SVM and fusion feature extraction is shown in Figure 3.

The pseudocode for the wind turbine blade fault detection method based on TROA-SVM is shown in Algorithm 1.
**Algorithm 1**: TROA-SVMInputs: dataset, *D*; population size, *N*; maximum number of iterations, max_iter; inertia weights, *w*; acceleration constants, *C*1, *C*2; perturbation intensity, *α*.Output: optimized SVM parameters (C*,gamma*).1.
$\;\mathrm{Initialize}\;\mathrm{the}\;\mathrm{positions}\;\mathrm{of}\;\mathrm{the}\;\mathrm{Tyrannosaurus}\;\mathrm{Rex}\;\mathrm{population},\;X_{i}$$,\;\mathrm{where}\;\mathrm{each}\;\mathrm{position}\;\mathrm{represents}\;a\;\mathrm{parameter}\;\mathrm{vector}\;(C_{i},{\mathrm{gamma}}_{i})$.For *i* = 1 to *N* use    
$C_{i}$ = rand(0, 1)    
${\mathrm{gamma}}_{i}$ = rand(0, 1)end for2. For iteration *t* = 1 to max_iter, use3. For each individual Tyrannosaurus Rex *i* = 1 to *N* use4.  
$\mathrm{Calculating}\;\mathrm{individual}\;\mathrm{adaptation}\;F(X_{i})$
5.  
$\mathrm{if}\;F(X_{i})$$\;\text{>}F(P_{best})$ then6.   
$P_{best}$
$=X_{i}$
7.  end if8.   end for9. 
$\mathrm{Determination}\;\mathrm{of}\;\mathrm{global}\;\mathrm{optimum}\;\mathrm{position}\;G_{best}$
10.  
$\mathrm{If}\;F(P_{best})$$\;\text{>}F(G_{best})$ then11.  
$G_{best}$
$=P_{best}$
12. end if13. For each individual Tyrannosaurus Rex *i* = 1 to *N* use14.  
$\mathrm{The}\;\mathrm{rate}\;\mathrm{of}\;\mathrm{renewal}\;\mathrm{of}\;\mathrm{individuals}\;V_{i}(t+1)$:15.   
$V_{i}(t+1)=w\cdot V_{i}(t)+c_{1}\cdot rand\cdot (P_{best}-X_{i}(t))+c_{2}\cdot rand\cdot (G_{best}-X_{i}(t))$
16.   
$\mathrm{Update}\;\mathrm{the}\;\mathrm{location}\;\mathrm{of}\;\mathrm{individuals}\;\mathrm{at}\;X_{i}(t+1)$:17.   
$X_{i}(t+1)=X_{i}(t)+V_{i}(t+1)$
18.    Randomized perturbations to avoid local optimization:19.   
$X_{i}(t+1)=X_{i}(t+1)+\alpha \cdot randn$
20.  
$\mathrm{Ensure}\;\mathrm{that}\;\mathrm{the}\;\mathrm{parameter}\;X_{i}(t+1)$ is in the range [0, 1].21. end for22. Check that the maximum number of iterations has been reached or that the classification accuracy is satisfactory23. if conditions are met then24.  break25. end if26. end for27.
$\mathrm{Output}\;\mathrm{the}\;\mathrm{global}\;\mathrm{optimal}\;\mathrm{position}\;G_{best}$
 as optimized SVM parameter (C*,gamma*)


## 3. Results

### 3.1. Experimental Platforms

The experimental platform mainly includes sound sensors, voiceprint monitoring equipment, local area network switches, data storage and analysis servers, cloud server platforms, and other equipment. Due to the strong interference at the wind turbine site, the sound pickup device is arranged around the wind turbine tower, and according to the characteristics of the wind turbine and the need for acoustic acquisition, the sensor is erected on the downwind side (i.e., the direction of wind avoidance when the blades are rotating) of the wind turbine blades through the bracket, and is kept at least 3.5 m from the center of the tower with a mounting height higher than 2.1 m in order to obtain the optimal signal-to-noise ratio. Considering that the blade rotation plane will change with the wind direction, it is necessary to use multiple pickup sensors for signal acquisition to ensure the comprehensiveness and accuracy of the data. The pickup sensor is wrapped with a sponge to reduce the interference of ambient noise, so as to obtain a higher quality measurement signal. When the acquired raw signal first enters the signal conditioning module, the signal conditioning module filters and amplifies the acquired noise signal to improve the signal-to-noise ratio. The conditioned signal is then digitized by the data acquisition card. These digitized data are transmitted to the analysis and storage server through a high-speed communication link for later data processing and analysis. The diagram of fault detection experimental platform for the wind turbine blade is shown in Figure 4.

The wind turbine blade sound signal acquisition device, i.e., the voiceprint monitoring device, is shown in Figure 5, where the sound sensor is a HY205 microphone (Hunan Shengyi Measurement & Control Technology Co., Ltd, China) with a performance level of SJ/T10724-2013 Level 2 and a nominal sensitivity of 50 MV/Pa., from Hunan Shengyi Measurement & Control Technology Co., Ltd, China. Wind turbine blades are constructed of composite materials with a tapered blade geometry that tapers from root to tip. The root of the blade is thicker and stronger to withstand the forces generated by the turbine hub, while the tip is thinner to reduce drag and noise. The aerodynamic profile of the blades is designed to resemble an aircraft wing to optimize efficiency and reduce drag. Blade lengths range from 40 to 60 m. The voiceprint monitoring device includes sound sensors, signal conditioning modules, data acquisition cards, and network communication interfaces.

### 3.2. Feature Extraction

After acquiring the sound signals generated by the rotation of wind turbine blades, as shown in Figure 6, normal vs. icing time–frequency diagrams, the multi-scale feature extraction algorithm proposed in Section 2 was used to extract acoustic signal features. Due to multidimensional features in each frame of the signal, the sensitivity of each dimension may be different. Through comparative analysis, it was found that among the extracted 13 dimensional MFCC features, the second, third, and sixth dimensional features were more effective in distinguishing between faulty and normal samples. Therefore, these three features were selected as the feature vectors for MFCC. In order to further enhance the classification ability of the model, a six-dimensional feature matrix is constructed by combining other important audio features, including short-term energy, short-term average zero crossing rate, and short-term power spectrum, to more comprehensively capture key features in the acoustic signal and improve the accuracy of fault detection. The distribution of these six features is shown in Figure 7.

### 3.3. Model Training

When training the fault detection model, normal samples were labeled as ‘0’, and fault samples were labeled as ‘1’. The extracted six dimensional features (i.e., short-time energy, short-time average zero crossing rate, short-time power spectrum, and the second, the third, and the sixth dimensional features of MFCC) were used as network inputs. The parameters of the TROA-SVM algorithm were as follows: the number of populations was 30, the maximum number of iterations was 50, the number of parameter dimensions was 2, the lower bound of the search space was [0.01, 0.01], and the upper bound of the search space was [10, 10]. In SVM, *C* = 1.0 and gamma = 5.0. In order to evaluate the impact of different parameters on the SVM’s performance, the SVM model with multiple sets of parameters was compared, and the accuracy, the recall rate, and the F1_score of the models with different of *C* and gamma are shown in Table 1.

Table 1 presents model evaluation metrics for different hyperparameter combinations of the support vector machine (SVM). It shows that when the hyperparameters were *C* = 3.585 and gamma = 6.703, the model achieved the highest performance, with an accuracy of 98.7%, a recall rate of 97.9%, and an F1 score of 97.7%. Other hyperparameter settings, such as *C* = 1.0 and gamma = 0.5, also showed strong performances, but grid search and randomized search methods resulted in slightly lower accuracy and recall. The grid search method provided a combination that yielded an accuracy of 97.5%, while randomized search resulted in an accuracy of 97.0%.

When the parameter was optimized by TROA algorithm, *C* = 3.585 and γ = 6.703, which were better than those of model with other parameters. The confusion matrix for classifying normal and fault samples when *C* = 3.585 and γ = 6.703 is shown in Figure 7, and the detection results are presented in Table 2. In this experiment, the normal test sample was 3352 and the fault test sample was 2439. According to Figure 8 and Table 2, the accuracy of normal sample detection results was 99%, and the accuracy of fault sample detection results was 97%, which confirms the effectiveness of the TROR-SVM method.

To further compare the detection performance, t-SNE was used to visualize the diagnostic results (classification results), which is shown in Figure 9. In the figure, the yellow dots represent normal category ‘0’, and the blue dots represent the fault category ‘1’. As shown in Figure 9, although the fault samples and normal samples could be effectively separated, there are still cases wherein a small number of samples were incorrectly identified, i.e., a very small number of fault samples (blue dots) were erroneously detected as normal samples (yellow dots). Similarly, a very small number of normal samples (yellow dots) were mistakenly identified as faulty samples (blue dots). However, this classification

### 3.4. Fault Detection

When the wind turbine blades are faulty or normal, 60 s of blade sound signals are acquired separately. The experimental conditions were characterized by average wind speeds typically ranging from 5 to 15 m/s, with gusts up to 20 to 25 m/s. The wind speeds were generally in the range of 5 to 15 m/s. Temperatures were around −20 °C. In addition, blade surface temperatures could drop below −10 °C. The feature extraction method proposed in the article is used to obtain feature vectors, and 2000 fault samples (negative samples) and 3000 normal samples (positive samples) were input as test sets into different classification models (SVM, KNN, Adaboost, random forest) for detection. The fault detection results are shown in Table 3. In SVM, a linear kernel function was selected, and the penalty parameter, *C*, was 1.0. In K-Nearest Neighbors (KNN), *k* = 5, and Euclidean distance was chosen as the distance metric. In Adaboost, the maximum depth of the tree was set to 1. The random forest model was composed of 100 decision trees, and each maximum tree depth was 10.

According to Table 3, the positive sample detection accuracy of the TROA-SVM-based model is higher than 99%, and the negative sample detection accuracy is higher than 97%, which has the best detection effect and significant advantages in wind turbine blade fault state monitoring.

## 4. Conclusions

(1)In response to the limitations of traditional wind turbine blade fault detection techniques based on images, vibrations, etc., this paper proposes a wind turbine blade fault state detection method based on acoustic features, with an average accuracy of over 97%.(2)A wind turbine blade fault detection method based on TROA-SVM was proposed, and it was compared with SVM, KNN, Adaboost, and random forest. The experimental results confirmed that the TROA-SVM method has higher detection accuracy.

Although the method is effective in troubleshooting wind turbine blades, it can still be improved and future plans are as follows:(1)Expansion of Fault Types: In addition to typical blade icing conditions, future work could explore the detection of other fault types. Incorporating multiple fault conditions, such as blade breakage, into the system will enhance the versatility and applicability of the system.(2)Enhancement of Environmental Adaptability: Although the current method has demonstrated strong adaptability to environmental conditions, factors such as temperature, humidity, and noise levels in different wind farms may still impact the fault detection performance. Future research can focus on optimizing signal preprocessing techniques and feature extraction processes to enhance the system’s robustness against variations in environmental factors.(3)Multi-dimensional Feature Fusion for Fault Detection: Given that wind turbine blade faults may involve multiple fault types, future research could explore the fusion of various signal sources, such as acoustics, vibrations, and temperature, to improve the accuracy and reliability of fault detection. For instance, deep learning models can be employed to integrate multi-dimensional features, providing a more comprehensive approach to fault identification.

## Figures and Tables

**Figure 1 sensors-25-00720-f001:**
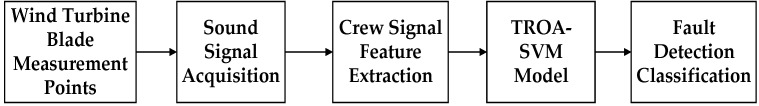
Fault detection for wind turbine based on TROA-SVM.

**Figure 2 sensors-25-00720-f002:**
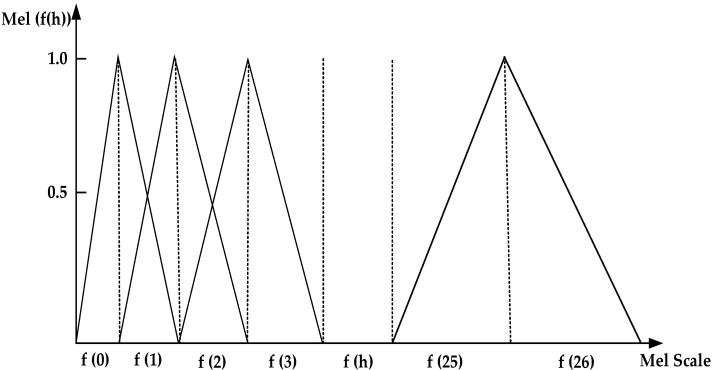
MFCC triangular filters.

**Figure 3 sensors-25-00720-f003:**
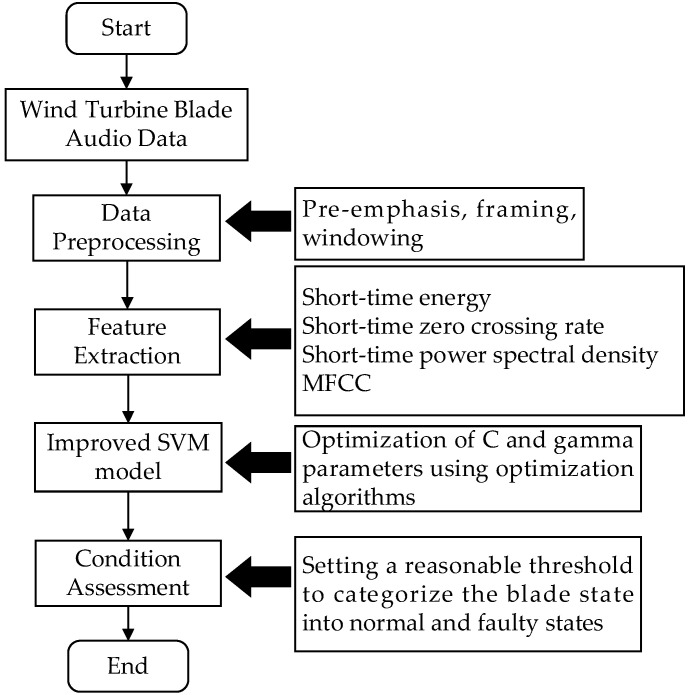
The fault detection algorithm based on TROA-SVM and feature extraction.

**Figure 4 sensors-25-00720-f004:**
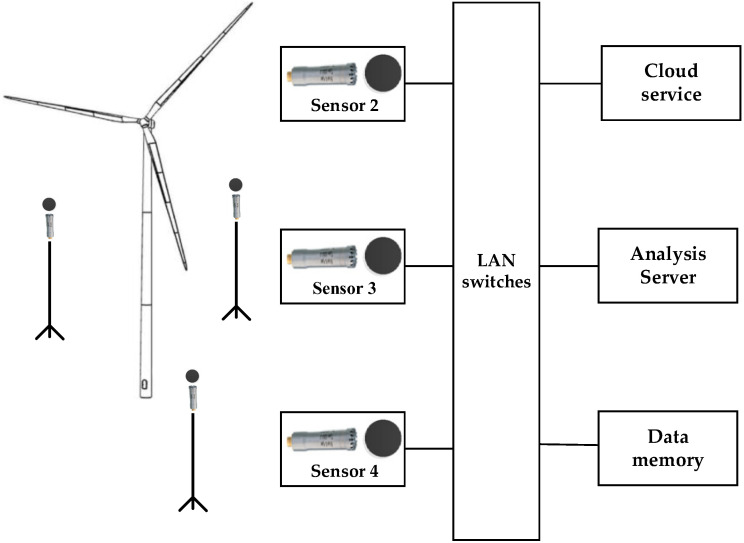
Fault detection experimental platform of wind turbine blades.

**Figure 5 sensors-25-00720-f005:**
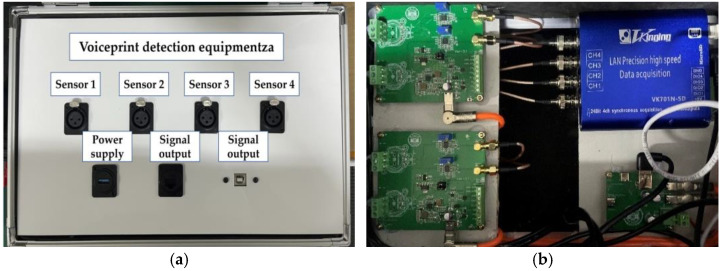
Wind turbine blade sound signal acquisition device. (**a**) Voiceprint monitoring device, (**b**) internal circuit for voiceprint monitoring device.

**Figure 6 sensors-25-00720-f006:**
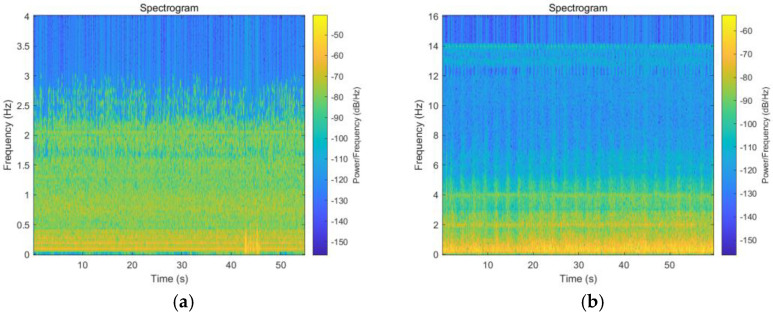
Icing vs. normal time—frequency diagrams. (**a**) Ice time—frequency diagrams; (**b**) normal time—frequency diagrams.

**Figure 7 sensors-25-00720-f007:**
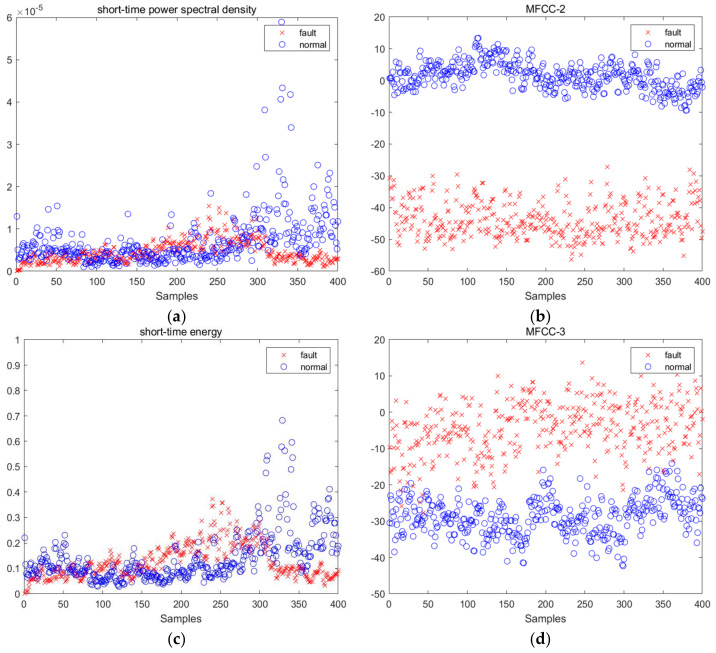
Feature distributions of normal and faulty samples. (**a**) Short—time power spectral density; (**b**) the 2nd MFCC; (**c**) short—time energy; (**d**) the 3rd MFCC; (**e**) short—time zero crossing rate; (**f**) the 3rd MFCC.

**Figure 8 sensors-25-00720-f008:**
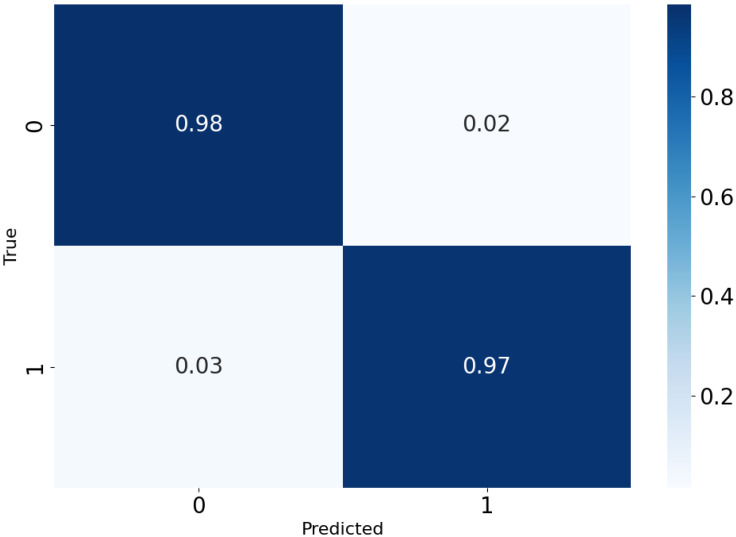
Confusion matrix for classifying normal and fault samples.

**Figure 9 sensors-25-00720-f009:**
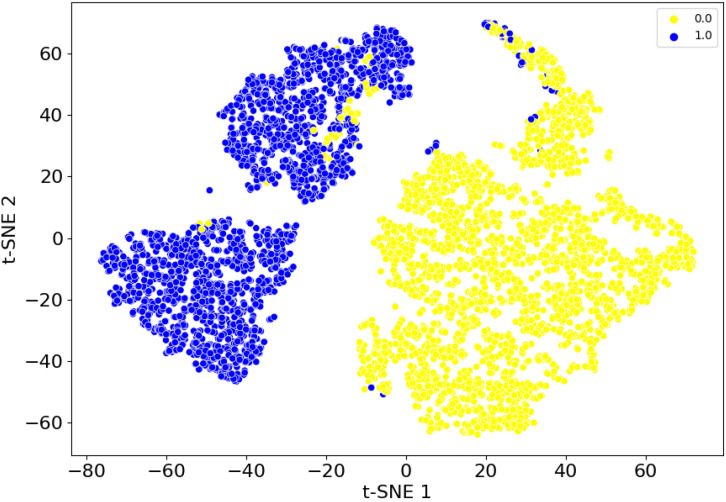
Data visualization of diagnostic results.

**Table 1 sensors-25-00720-t001:** Comparison of different parameter combinations.

(*C*, gamma)	Accuracy	Recall Rate	F1_Score
(1.0, 0.5)	96.5	94.8	95.0
(3.585, 6.703)	98.7	97.9	97.7
Grid search is optimal	97.5	96.2	96.5
Randomized search for optimal	97.0	95.5	95.8

**Table 2 sensors-25-00720-t002:** Detection results of wind turbine blade with the TROA-SVM model.

Tags	Accuracy	Recall Rate	F1_Score	Test Sample	Wrong Scores
0	0.99	0.99	0.98	3319	33
1	0.97	0.97	0.97	2368	71

**Table 3 sensors-25-00720-t003:** Results of different fault detection methods.

Algorithm	Negative Sample Accuracy	Positive Sample Accuracy
SVM	93.76	94.82
KNN	94.91	94.07
Adaboost	86.23	87.10
Random Forest	95.07	96.23

## Data Availability

Data are unavailable due to privacy or ethical restrictions.

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
