# Peer review of "Wind Turbine Blade Fault Detection Method Based on TROA-SVM"

_sensors, 2025, doi:10.3390/s25030720_

Round 1

Reviewer 1 Report

Comments and Suggestions for Authors

This paper developed an improved method, which integrates the Tyrannosaurus Optimization Algorithm (TROA) with Support Vector Machine (SVM), aiming to enhance the diagnosis accuracy of wind turbine blade. Overall, this paper is well-structured and presents some promising results. Here are some suggestions for further improving the quality of this paper:

(1)    Please give the reasons or the contributions for choosing the Tyrannosaurus Optimization Algorithm (TROA) with Support Vector Machine (SVM) in Introduction.

(2)    Is the title of the section 2.4 appropriate? It is recommended to add textual explanations instead of just one table.

(3)    Please adjust the text in Figure 4, for example: Word “Acquisition” is divided into two lines.

(4)    Please enrich the conclusion content and suggest adding future research plans.

(5)    The resolutions of the figures should be improved. For example, in Fig. 1.

--------------

Supplement

• What is the main question addressed by the research?

it integrates the Tyrannosaurus Optimization Algorithm (TROA) with Support Vector Machine (SVM), aiming to enhance the accuracy and reliability of fault detection about wind turbine blade.

• Do you consider the topic original or relevant to the field? Does it address a specific gap in the field? Please also explain why this is/ is not the case. 

the topic  relevant to the field; it address a specific gap in the field.

the reason is that : After experimental comparison, the proposed algorithm has good diagnostic accuracy, providing a new method for fault diagnosis based on acoustic signals.

• What does it add to the subject area compared with other published material?

A novel method is proposed to perform the fault diagnosis of turbine blade.

• Are the conclusions consistent with the evidence and arguments presented and do they address the main question posed? Please also explain why this is/is not the case.

I have given the suggestions for the conclusion.

• Are the references appropriate?

Yes

• Any additional comments on the tables and figures.  

I have given the suggestions for  figures.  

Comments on the Quality of English Language

(1)    There are some grammar errors in this manuscript. Please check the whole manuscript and address these kinds of issues throughout the whole manuscript.

Reviewer 2 Report

Comments and Suggestions for Authors

The reviewed article discusses the topic of "Wind Turbine Blade Fault Detection Method Based on TROA-SVM." In the opinion of the reviewer, the article is interesting; however:

1) Figure 3 describes the fault detection algorithm. Should we parameterize the individual feature extraction methods each time during the feature extraction step?

2) What is the required amount of data to meet the assumptions of the proposed algorithm?

3) Figure 6 exceeds the margins of the article.

4) Table 1 extends beyond the margins of the article. The same applies to the other tables.

5) The conclusion section needs to be expanded.

6) The literature cited is biased.

7) The placement of measurement sensors is debatable. What happens in cases of strong winds, nearby turbines, or other sources of noise? What are the recommendations for conducting measurements considering the impact of the external environment? During noise measurements, factors such as temperature, pressure, wind speed and direction, humidity, altitude, and the angle of sensor placement are crucial. The conducted experiment is highly debatable.

Reviewer 3 Report

Comments and Suggestions for Authors

The authors don't give any information about the blades such as the geometry of blade, the length of blade etc.

The authors don't give any data of tests in real conditions such as wind speed, temperature etc. which are important data in fault detections.

Figure 3, fault algorithm flowchart can't work correctly because the flowchart must be in a closed loop.

The figure 6 (Feature distributions of normal and faulty samples) don't give a comforting conclusion because of the large difference in results.

The references are poor.
The authors can use the two references below to improve their paper:

-A smart multiphysics approach for wind turbines design in industry 5.0, ELSEVIER, Journal of Industrial Information Integration, 2024.

-Wind Turbine Blade Fault Detection via Thermal Imaging Using Deep Learning, IEEE-2024 Intermountain Engineering, Technology and Computing (IETC).

Comments on the Quality of English Language

Some modifications and rewording are necessary.

Round 2

Reviewer 2 Report

Comments and Suggestions for Authors

The answers to my questions are satisfactory. The article can be published in the present form.

Author Response

Dear Editor,

Thank you for your feedback! I'm glad the answers meet your expectations. I appreciate your confirmation that the article is ready for publication in its current form.

Sincerely yours,

Haijun Lin

Reviewer 3 Report

Comments and Suggestions for Authors

The authors have answered all of my questions but the authors can improve the reference part. Above all, they can add some references about different fault detection methods in literature.
The authors can use this reference: Fault diagnosis of smart grids based on deep learning approac, IEEE- 2021 World Automation Congress (WAC)

Author Response

Dear Editor,

Thank you for pointing this out. We have updated and supplemented the references in the revised manuscript.

Sincerely yours,

Haijun Lin
